# Automated Diagnosis of Diabetic Retinopathy Using Deep Learning: On the Search of Segmented Retinal Blood Vessel Images for Better Performance

**DOI:** 10.3390/bioengineering10040413

**Published:** 2023-03-26

**Authors:** Mohammad B. Khan, Mohiuddin Ahmad, Shamshul B. Yaakob, Rahat Shahrior, Mohd A. Rashid, Hiroki Higa

**Affiliations:** 1Department of Biomedical Engineering, Khulna University of Engineering and Technology, Khulna 9203, Bangladesh; 2Department of Electrical and Electronic Engineering, Khulna Engineering and Technology, Khulna 9203, Bangladesh; 3Faculty of Electrical Engineering and Technology, Universiti Malaysia Perlis, Arau 02600, Malaysia; 4Department of EEE, Noakhali Science and Technology University, Noakhali 3814, Bangladesh; 5Department of Electrical and Systems Engineering, University of the Ryukyus, Okinawa 903-0129, Japan

**Keywords:** deep learning, retinal blood vessels, diabetic retinopathy, segmentation, convolutional neural networks

## Abstract

Diabetic retinopathy is one of the most significant retinal diseases that can lead to blindness. As a result, it is critical to receive a prompt diagnosis of the disease. Manual screening can result in misdiagnosis due to human error and limited human capability. In such cases, using a deep learning-based automated diagnosis of the disease could aid in early detection and treatment. In deep learning-based analysis, the original and segmented blood vessels are typically used for diagnosis. However, it is still unclear which approach is superior. In this study, a comparison of two deep learning approaches (Inception v3 and DenseNet-121) was performed on two different datasets of colored images and segmented images. The study’s findings revealed that the accuracy for original images on both Inception v3 and DenseNet-121 equaled 0.8 or higher, whereas the segmented retinal blood vessels under both approaches provided an accuracy of just greater than 0.6, demonstrating that the segmented vessels do not add much utility to the deep learning-based analysis. The study’s findings show that the original-colored images are more significant in diagnosing retinopathy than the extracted retinal blood vessels.

## 1. Introduction

Diabetic retinopathy (DR) has been wreaking havoc on the modern-day population given the estimation that 463 million of the global population in 2019 and 700 million in 2045 are affected by diabetes mellitus (DM). Diabetic retinopathy appears to be a prevalent consequence of DM, thus being one of the major causes of blindness in the working population, as suggested by the International Diabetes Federation (IDF) [1]. Due to the lack of precise diagnosis, leaving DR untreated at the following stages can cause blindness; it has been responsible for 5 percent of the global blindness being diagnosed, where estimations suggest 50–65 cases of blindness occur for every 100,000 [2].

Biomarkers behind biological and pathological processes of diabetic retinopathy, such as blood pressure, diabetes duration, glucose level, and cholesterol levels, are considered to be unquestionably important determinants of the development of this disease, although growth of DR cannot be determined from these [3]. This is in contrast to patients with poor control and strong glycemic control, which can suddenly deteriorate and can allow for the identification of various phenotypes of progression of DR following the nature of the retinal lesions.

A comprehensive examination must take place for the proper evaluation of this disease. Likely, all patients suffering from type 1 diabetes and approximately 60% from type 2 diabetes will develop a particular degree of DR within 20 years of diagnosis [4]. Thus, regular screening is a prerequisite for catching the disease at an early stage.

Consequently, a precise diagnosis of DR must be a prerequisite in the case of addressing a patient regarding the compatible treatment, as DR treatments are vastly driven according to their severity levels. In these terms, deep learning-based image processing can accelerate the DR classification proficiency on the basis of precise recognition of its severity, which can provide accurate prediction of patient’s morbidities as well as ensure enriched diagnosis and, hence, can aid in designing plausible treatment plans for the cure. Image processing has been enormously engaged in DR classification by highlighting where fundus cameras are employed to collect retinal fundus images. Techniques including image enhancement, fusion, morphology identification, and image segmentation give a rise to the cognitive efforts of medical physicians in extracting additional information from medical image data [5]. Several attempts have been found in the case of automated DR classification using DL, where proposed methods encompass the types of categorizations based on lesions and blood vessels [6], as detailed in the following sections.

### 1.1. State of the Art on Approaching DR Detection with Deep Learning Techniques

Studies have been extensively carried out pertinent to diabetic retinopathy, suggesting multiple techniques proven adept for DR detection. This section represents the deep learning and neural network technique approaches for the multiclass classification of diabetic retinopathy. A novel automated recognition system was developed by Abbas et al. [7] under the five severity levels of diabetic retinopathy where pre- or post-processing of fundus images was not required. A semi-supervised deep learning technique was utilized in tandem with fine-tuning steps, where the sensitivity of 92.18% and specificity of 94.50% were obtained.

In another study, diabetic retinopathy detection was carried out using fundus images from the EyePACS1 dataset, containing approximately 9963 images that were collected from 4997 patients, and the Messidor-2 dataset, which contained 1748 images acquired from 874 patients. For detecting RDR using operation cut points of high specificity, the sensitivity and specificity gained for EyePACS-1 were 90.3 and 98.1 percent, respectively. On the other hand, 87.0% and 98.5% were acquired in terms of sensitivity and specificity, respectively, for the Messidor-2 dataset [8]. The architecture used was the Inception v3 [9], which showed impressively high sensitivity and specificity. DR severity grading was performed, yielding 93.33% accuracy by applying a quadrant-based approach; here, [10,11] addressed the issue of scarcity of annotated data for training purposes. The model used was Inception v3. The model training was conducted on a subsample of a Kaggle diabetic retinopathy dataset, while the accuracy testing was carried out on another subset of data. Transfer learning was implemented. In [12], the detection was shifted onto different platforms, such as smartphones and smartwatches. Deploying automated DR detection on such ubiquitous platforms, cost-effectively provides healthcare, offering a frictionless healthcare system. A convolutional neural network (CNN) model was based on Inception, which also serves as an ensemble of classifiers and also functions as a binary decision-tree-based method. Goncalves et al. compared human graders and the agreement of different machine learning models. Regardless of the dataset, transfer learning has performed well in terms of agreement across different CNNs.

A comparison had been undertaken between traditional approaches and CNN-based approaches. The Inception v3 model has been nonpareil, having reached the accuracy of 89% on the EyePACS dataset and performing the best [13]. In another study, the fundus images were classified into average to extreme conditions versus non-proliferative DR [14], where they used backpropagation neural organization (BPNN). Table 1 highlights the multiple deep learning approaches for DR detection and classification over various datasets of retinal fundus images.

### 1.2. Research Gap

Patients at higher risk in the proliferating group should be addressed for prompt remedy and diagnosis, which demands the diagnostic technique to be highly precise and appropriate, in short, serves as an urgent call for a proficient and self-contained feasible approach for retinopathy identification, thus providing reliable results. Thus, we have seen pre-trained CNNs and other deep learning techniques being used to classify multiple diseases in the past. While taking this view into account, a furnished dataset and deep transfer learning are required for the improvement of classification accuracy. Otherwise, the dataset comprising low-resolution DR images mentioned in previous sections, where research has taken on pre-trained and traditional approaches, may lead to erroneous classification followed by misleading accuracy. While considering the feasibility in the case of DR detection following deep learning-based analysis, both the original-colored and segmented images had been used for diagnosis earlier by researchers. Nonetheless, which approach is clinically efficient remains equivocally a matter of doubt. In short, it indicates that a mechanism for assessing the classification performance characteristics of modern deep learning approaches on relevant datasets should be enriched.

### 1.3. Seleciton of the Original Dataset and Derivation of the Segmented One for Our Study

In terms of DR classification, the size and quality of the obtained dataset vastly determine the classification accuracy, that is, a higher accuracy requires a large amount of training data using a deep learning algorithm. Thus, considering the quality assurance, the dataset should be gained from reliable sources with accurate tags. Here, in Table 1, we mention some of the datasets widely used for DR detection, including following the Kaggle Diabetic Retinopathy dataset [28,29], DiaretDB1 dataset [30], HRF (High Resolution Fundus Image database) [31], and the Messidor and Messidor-2 datasets [32].

One of the two datasets involved in this research is the HRF database [31] which is used to train our transfer learning model for segmentation; it comprises three sets of fundus images, including 15 images of healthy patients, 15 images of DR patients, and lastly, the same number of images of patients with glaucoma. Each image from three sections has binary gold standard vessel segmentation images of its own. A group of professionals from the field of retinal image analysis as well as the clinicians from the cooperated ophthalmology clinics contributed to generating these data. In addition, the masks illustrating the field of view (FOV) are provided for particular datasets. The plane resolution of HRF is 3504 × 2336, which is relatively high compared with other available datasets in this field, asserting it as the more worthy one for our segmentation purpose.

As a part of our study, another dataset we used is the APTOS Blindness Dataset provided by Kaggle [30] and generated by Aravind Eye Hospital located in India, whose original-colored and segmented blood vessels had been used to meet our research objectives. The goal was to derive the solutions from other ophthalmologists through the 4th Asia Pacific Tele-Ophthalmology Society (APTOS) Symposium. A large dataset of retinal images was generated using fundus photography, more precisely a photograph of the rear of the eye. The images were rated in the range 0–4 inclusive, where different ratings correlate to different stages of DR, except 0, which is to be assumed indicative of no symptoms of DR. One of the major reasons for choosing this one among multiple options, as illustrated in Table 1, is because of its greater size. It is the third-largest dataset, consisting of 5590 images, where the DR grading followed the ICDRDSS protocol and contained appropriate class distribution of images into each of the relative grades according to their severity levels. Figure 1 represents two of the original-colored images along with their segmented one to signify the dataset quality of this study. 

## 2. Proposed Methodology

### 2.1. Dataset Preprocessing and Enhancement

At the outset, the HRF dataset went through data augmentation. The U-Net network used this dataset as input for an initial transfer learning phase. Data augmentation uses certain techniques to artificially elevate the size of the data, meaning an overall quantitative augmentation. Deep learning models require ample training data, and a perennial problem is the shortage of training data, which is certainly the case in the medical image processing field. Techniques of augmenting data include position augmentation, such as scaling, flipping, cropping, padding, translation, affine transformation, rotation, and color augmentation, including changing contrast, saturation, and brightness, to make the images consistent in case of intensity and size that will contribute to CNN for precise classification.

In this preview, we implemented horizontal flipping, vertical flipping, and rotation to elevate our data. The point to note is that only training data went through augmentation. We initially had 45 total images. There was an 80:20 bifurcation of the data such that 36 of the images from that initial pool went through the augmenting procedures. This is intended to be training data. A fourfold increase in training data was achieved, meaning we now had an expanded pool of images which bodes well for the training of our model. This will essentially help the model generalize better. The latter portion of that split was used for validating the model. All images were converted to a size of 224 × 224 to be used for input on the convolutional network. Figure 2 shows the proposed methodology.

For the APTOS dataset, the data had to be run through a filtering process where we separated it into three portions: the training set, the validation set, and the test set. Initially, we had 3662 images of training data. Seventy percent of those images, that is, 2563 images, were used for the actual training of the model. The remaining 30 percent of the leftover images were bifurcated, meaning that from the 1099 images that were left untouched, 549 images were designated as one group and the other 550 images as another group. The former is used for validation, while the latter is used for testing. All the images were changed to a size of 224 × 224 before operation. Here, Ben Graham’s preprocessing was coupled with auto-cropping, which upgraded our training performance [33]. The images were scaled to a certain radius, and the local average color was subtracted. The need for this arose due to a lack of lighting in a section of the images. One workaround was to convert the images to greyscale, but the aforementioned preprocessing method was selected instead. Cropping is used to shave off the uninformative areas of images. Data augmentation techniques, specifically horizontal flipping and vertical flipping, were used for artificially enriching the data and ultimately boosting performance.

### 2.2. CNN Architectures and our Suggested Workflow

In recent years, deep neural networks based on CNN models have been vastly engaged to address the disease classification challenges; they are assisted by computer vision since CNN has appeared to be applied in the field of deep learning, computer vision, and medical image processing with collective success and progression. Some of the related works corresponding to renowned authors depict the range of work that has been carried out. In medical image processing, CNNs run the gamut from performing pneumonia detection using chest x-rays [34] to brain tumor detection using MRI scans [35]. The U-Net architecture has been extensively used for segmentation [36], for instance, for prostate zone segmentation [37].

The Inception v3 architecture has achieved performances near the level of humans, where tasks such as colorectal cancer lymph node metastasis classification [38] and skin cancer classification have fared well [39]. Under the umbrella of the DenseNet architecture, studies have been carried out on image classification [40], COVID-19 diagnosis [41], as well as many other studies.

Hence, we deliberately chose CNN models to handle the DR classification more precisely. Our proposed framework comprises two phases, including segmentation and classification. In the case of segmentation purposes, CNN-based segmentation was chosen over other image segmentation techniques. To be more precise, U-Net CNN architectures were engaged for DR severity classification in our study, which were later carried out to transfer learning, as discussed in Section 3. Another phase denoting classification refers to the employment of several CNN-based architectures following Inception v3 and DenseNet-121, which have boosted the chances for robust classification of DR severity.

Since the APTOS original dataset had no previous instance of being segmented, there is a prerequisite for this flow of work to segment the dataset. To meet our research goal, the U-Net model has been trained on the HRF dataset, which is later passed through a transfer learning phase and then applied on the APTOS Blindness dataset to derive the instance segmented dataset from the original one. Then, the pre-trained models (Inception v3 and DenseNet-121) performed classification tasks on the original-colored images of the APTOS Blindness dataset as well as on instance segmented blood vessels image dataset as mentioned in Figure 2. Finally, the robustness of CNN architecture’s classification performances on our original and instance segmented dataset were evaluated for the precise diagnosis of DR patients.

## 3. Pre-Trained CNN Architectures and Experimental Setups

### 3.1. Segmentation

The U-Net convolutional network is used for our segmentation phase. There is a multitude of image segmentation techniques: region-based image segmentation, edge-based segmentation, clustering-based image segmentation, and, of course, convolutional neural network (CNN)-based image segmentation [42]. CNN-based segmentation is on the cutting edge of this field of research. Extrapolating various regions in an image and demarcating those regions into different classes is image segmentation. In simple terms, an image is broken down for segmentation into multiple regions. The intention behind this is to make images more ‘palatable’, meaning representing images in a format suitable for analysis by machines. We segmented the images of the APTOS dataset using a U-Net CNN architecture later to be worked on by other CNNs for DR severity classification.

#### 3.1.1. U-Net

U-Net is a breakthrough architecture in medical image processing and is a successor to the sliding-window approach [43]. The sliding-window approach by [44] had downsides, as there was redundancy due to overlapping patches and a lack of cost-effectiveness [36]. U-Net architecture does more with less, as it is trained with fewer training images but provides comparatively more accurate segmentation. Fully convoluted networks (FCN) do not contain any dense layers, but this network extends FCNs that won the ISBI 2015 challenge [45].

#### 3.1.2. Transfer Learning with U-Net

The U-net has gone through a transfer learning phase. Taking inspiration from humankind, transfer learning refers to the phenomenon of transferring knowledge across different tasks. The chances of transfer of learning increase the more related the newer task is to the older one. In transfer learning, weights and features from earlier trained modules can be used for another task. Low-level features, including edges, intensity, and shapes, can be transferred across tasks denoting a transfer of knowledge. The U-Net model was trained on the HRF dataset, which had been augmented beforehand. After the training phase, the model was saved, and later the U-Net model was used on the APTOS Blindness dataset, resulting in an instance segmented version of the original images as the output.

### 3.2. Classification

CNNs are adept at reducing the number of parameters without losing the quality of the model. An image goes through an analysis where a multitude of image features are scrutinized, and the outputs are demarcated into separate categories. The convolutional networks used for classification purposes are Inception v3 and DenseNet-121.

To overcome the challenges of computational expense, over-fitting, and gradient updates, the Inception framework provides multiple sizes of the kernel on the same level, opting to go wider rather than deeper following a heavily engineered route; the latest Inception v3 is well-received, achieving good accuracy on the ImageNet dataset. Inception v3 is 48 layers deep. Modifications from earlier models include factorizing larger convolutions into smaller ones, and asymmetric convolutions following a 5 × 5 convolution are replaced by two 3 × 3 convolutions to reduce parameters.

DenseNet is a breed of CNN characterized by dense connections between layers, thus being preferred since deeper networks are more adept at better generalizing [46], as the depth allows the network to learn far more complex functions. The deeper the network, the more chance for input information to vanish, which is dubbed the vanishing gradient problem. Resolving this issue entails moving away from the quintessential CNN architecture and installing dense layers requiring fewer parameters [47], and each feature is passed through layer by layer, being concatenated at each stage. Bottleneck layers are embedded in the architecture. DenseNet-121 has a total of 120 convolutions with 4 AvgPool. To elucidate further, it has 1 7 × 7 convolution layer, 58 3 × 3 convolution layers, a total of 61 1 × 1 convolution layers, the aforementioned 4 AvgPool, and 1 fully connected layer. However, here instead of AvgPool, global average pooling was used.

### 3.3. Original and Segmented Image Classification

One segment of the workflow entailed using the pre-trained DenseNet-121 and Inception v3 on the original-colored APTOS dataset images. The intention was to classify the data as per the categorization scheme of DR mentioned before. Both pre-trained models were used on the original images and segmented versions of those of the colored images. Before that, the U-Net was trained previously on the HRF dataset, and then that trained model was applied on the original-colored images to derive the segmented versions of itself, where the accuracy achieved was extremely high, approximately 99.02%. This higher accuracy, thus, validates the significance and feasibility of our segmentation process more precisely. Moreover, the data were categorized according to the aforementioned stages of DR.

The following applies to both workflows mentioned above for the DenseNet-121 and Inception v3. Both models were warmed up, using a total of 10 epochs from weight initialization, and a total of 20 epochs were used for training. Table 2 summarizes the training parameters of InceptionV3 and DenseNet-121. 

Stochastic gradient descent (SGD) was used for optimization. Gradient descent is continued iteratively to determine the optimal values of parameters, initiating from a starting value which is intended to enumerate the minimal possibility of a given cost function. Three types of gradient descent are used: batch, mini-batch, and stochastic gradient descent. Among these three types, the stochastic gradient descent is best suited in terms of vast datasets, as quintessential optimization techniques of gradient descent following batch gradient descent are intended to pursue the entire dataset as a batch. However, this strategy does not fare well on a typical technique like this, as it would take the entire dataset and run it on each iteration, thus incurring a heavy expense. The drawback is that this tends to be noisier, but training time is a priority. Thus, global average pooling was used. The layer input and the pool size are identical, and the average pool is taken. The input feature map is partitioned into smaller patches, where by applying max operation, the maximum of each patch was computed. In addition, the global average pooling layer lessens the intermediate dimensionality.

To introduce non-linearity into the network, the non-differentiable rectified linear unit (ReLU) function was implemented. ReLU is an activation function related to a particular input, where certain outputs are activated with non-zero values while others with zero values are turned off. The softmax activation function produces values in the range (0–1) and so is suitable for the output layer. The softmax output layer was reduced to five probability points corresponding to the five levels of DR severity given in Equation (1).
$$Softmax(y=j\;|\;\theta^{i})=\frac{e^{\theta^{i}}}{\sum_{j=0}^{k}e^{\theta }k^{i}}$$
where
$$\theta =w_{0}f_{0}+w_{1}f_{1}+\ldots .+w_{k}f_{k}=\sum_{i=0}^{k}w_{i}f_{i}=W^{T}F$$

Each input value is normalized into a vector of values. These values belong to a probability distribution. Here, *θ* is a one-hot encoded matrix which is a representation of categorical variables as binary vectors. This function predicts whether a set of features *f* are a class of *j*. Ultimately, the output is the ratio of the exponential of the input parameter and the sum of parameters of all existing values. A variable learning rate was used. The learning rate is a tuning parameter that determines the step size at each iteration. It dictates the level of change a model should go through in response to the predicted error when the weights are updated. Whenever the outputs stagnate for a given number of training epochs (i.e., hit a plateau), the learning rate is manipulated. Categorical cross-entropy was used as the loss function. Cross-entropy, in general terms, is a continuous and differentiable function that provides feedback necessary for steady incremental improvements in the model. This loss function, as given in Equation (2), is used when an example can only belong to one class out of the possible classes available, which is appropriate for our task at hand.
$$Loss=-\sum_{i=1}^{output\;size}y_{i}\times \log\bar{y_{i}}$$

Here, $\bar{y_{i}}=i^{th}$ scalar value for the output of the model, and $y_{i}=$ target value and denotes the probability that event *i* occurs. In our work, the cross-entropy loss between labels and predictions were calculated. In particular, we opted for this loss function, as we had more than two label classes.

## 4. Experimental Results and Performance Matrices

To reiterate the main aspects of our study, this is essentially a comparative study on original-colored images versus segmented images for DR severity levels detection.

### 4.1. Training and Validation Performance

From Figure 3, it can be seen that training accuracies were greater than validation accuracies for both the models applied on original and segmented blood vessels images. In contrast, the reverse was observed for loss since validation losses exceeded training losses. For both Inception v3 and DenseNet-121, the maximum validation accuracy and the minimum validation loss were attained while using the original images. A conspicuous trend was observed when comparing equivalent portions of the original and segmented results, which showed a preference for the original images over the segmented blood vessels images, as they offered an advantage to the classification performance.

### 4.2. Test Performance

Table 3 represents four performance metrics for each model applied on both original-colored and segmented images for gauging test performance. Looking at each class, the maximum values for all metrics achieved for both models occurred in the case of original-colored images compared with the segmented ones. For instance, the maximum values of precision, recall, and F_1_-score were found to be 0.97 each for Inception v3, whereas the maximum values of those three performance metrics were found to be 0.89, 0.93, and 0.91, respectively, for DenseNet121 for the No DR stage. Both were acquired in terms of original-colored images; these seemed to outperform the model’s performance metrics when applied to the segmented images since the maximum values of precision, recall, and F_1_-score, in this case, were 0.89, 0.93, and 0.91 for Inception v3 and 0.84, 0.96, and 0.9 for the DenseNet-121 model, respectively. This underlying pattern of better performances using original-colored images rather than segmented images was carried over to each class of severity levels.

In the same vein as Table 3, the data of the table were visualized in Figure 4, illustrating the same pattern of better metrics for both models on original images compared with those from segmented images.

The original images continued to follow the underlying pattern of better performance over segmented images regarding accuracy metrics. As shown in Figure 5, original images outscored the segmented images, attaining accuracies of 80% and 83%, respectively, compared with 72% and 69%, respectively, on segmented image classification. Regarding the context, the mean performance in terms of segmented images was between 0.6 and 0.7, which was outperformed, as previously, by that of the original image, which was 0.8. The values for Table 3 are pictorially depicted in Figure 6. Regarding the two dotted lines in the three tables, one represents the values from the original images, while the other represents the values from the segmented ones. Vertical and horizontal axes represent the actual values of the metrics and the severity levels, respectively. For the three charts, the blue bar remains relatively more elevated than the orange bar, and for every point, this difference stayed constant, implying better outcomes found for original images.

### 4.3. Comparison of Original and Segmented Datasets

Three different performance metrics, i.e., the F_1_-score, precision, and recall/sensitivity/TPR, were tabulated and are shown in Table 4. Values were found for all three metrics for the original images and segmented blood vessels with respect to all the five severity levels. The same underlying pattern of better performances using original images was found for all of the metrics, such as precision for original and segmented blood vessels for detecting No DR, which was 0.965 and 0.865, respectively. Likewise, the precision for Proliferative DR was 0.775 and 0.585, respectively, for the two models. For recall/sensitivity/TPR, the values for original and segmented blood vessels for detecting No DR were 0.975 and 0.945 and for detecting Proliferative DR, they were 0.465 and 0.15, respectively.

## 5. Discussion

Firstly, our preference for engaging deep learning models, including Inception v3 and DenseNet-121, is proven significant because of their evident aptitude in diagnosing medical images for disease identification. In terms of medical image diagnosis, Inception v3 provides the ability to adopt both global and local features from an image using different sized filters of convolution layers and pooling operations. In contrast, the dense connectivity pattern of DenseNet-121 ensures efficient retrieval and extraction of better features, leading to an improved outcome. The Resnet model is not included here because of its slightly poorer results compared with the other two included in this study. The performance metrics from the outcome show that the precision, recall, and F_1_-score obtained after the classification of original-colored images provides better performance than that for segmented blood vessels, for both state-of-the-art deep learning models. This indicates that the segmentation does not add much value to the diagnosis of diabetic retinopathy. The reason behind the lower performance using the segmented blood vessels can be explained in two ways. First, when we use the segmented blood vessels, the retinal blood vessels are extracted, while the colored image is converted to greyscale images, and the region outside the retinal blood vessel is filtered out. As a result of down-sampling in the segmentation process, some information (pixel values) from the retinal blood vessels is lost due to segmentation [48]. In addition, the region outside the vessel can contain important information (such as drusen and other biomarkers) which are essential biomarkers in a glaucoma diagnosis. Again, the loss of retinal lesions can be considered as a significant drawback in the case of using segmented images for diagnosis of DR.

Second, a certain type of disease shows different characteristics in retinal vascular structure [49]. If the changes in retinal vascular tissue in diabetic retinopathy are not as predominant as optic nerve disease, glaucoma, etc., then the segmented image will not add much utility to the diagnostic performance [50]. Furthermore, the confounding issue and overlap with other disease biomarkers in the retinal blood vessel will significantly impact the diagnostic performance.

Third, some studies have shown promise in using segmented drusen for the diagnosis of diabetic retinopathy [51]. Given that this study focuses only on retinal blood vessels and original images, segmented drusen are beyond the scope of the study. However, future research can be performed comparing the performance for original image and segmented images.

Lastly, the segmented blood vessels used in this study are the generated images from a trained model, which is trained on another dataset. The model trained with other images could have less ability to extract the retinal blood vessels from images of another dataset. This is likely due to different image acquisition devices and the quality of the data acquisition. The main finding from this study is that the original-colored images are better than the segmented blood vessels in the deep learning-based diagnosis of diabetic retinopathy. These findings support some of the previous studies, which used original fundus images for the classification purpose using deep learning. However, some of the studies claimed higher performance in classification using segmented blood vessels. The probable reason could be the segmentation model trained on a portion of their dataset (after manual annotation). Another reason could be that they used the original images in the segmentation without performing any under-sampling, which results in less information to lose. Therefore, using the high-resolution images could help to obtain high-resolution segmented blood vessels, which further helps achieve higher diagnostic performance since the analysis of the segmented image provides less resolution and precision compared with the analysis of original image. This finding can largely supplement the automated screening and real-time diagnosis of DR in clinical practices.

This study has some limitations which are worth mentioning. First, the image segmentation model was based on transfer learning, i.e., the original model was trained on a different dataset, and the images used in this study were used as a test dataset. Second, the study focuses only on disease classification or diagnosis. However, temporal analysis is required to track the disease progression, and the utility of the two approaches (original and segmented) also needs to be investigated. Future research could be performed on this aspect.

## 6. Conclusions

Diabetic retinopathy is one of the major retinal diseases impacting the human eye as well as causing blindness. Thus, early diagnosis is crucial, but manual diagnosis and limited human capability can lead to misdiagnosis. Therefore, obtaining a deep learning-based automated diagnosis of the disease could assist in feasible detection for treatment. In the case of deep learning-based analysis, it is still equivocal and unclear to choose between original and segmented blood vessels for diagnosis. In this study, a comparative analysis was conducted involving two different deep learning algorithms in two different approaches: colored images and segmented blood vessels. From the findings of this study, the segmented blood vessels were shown not to add much utility to the deep learning-based analysis. Hence, for diagnostic purposes, using the original images could help lessen the time and cognitive efforts of manual annotation and segmentation. As for future research, it is suggested to use temporal data for observing the contribution of the two different approaches in disease detection and progression as well as to derive a lesion-based classification for the precise understanding of DR severity level.

## Figures and Tables

**Figure 1 bioengineering-10-00413-f001:**
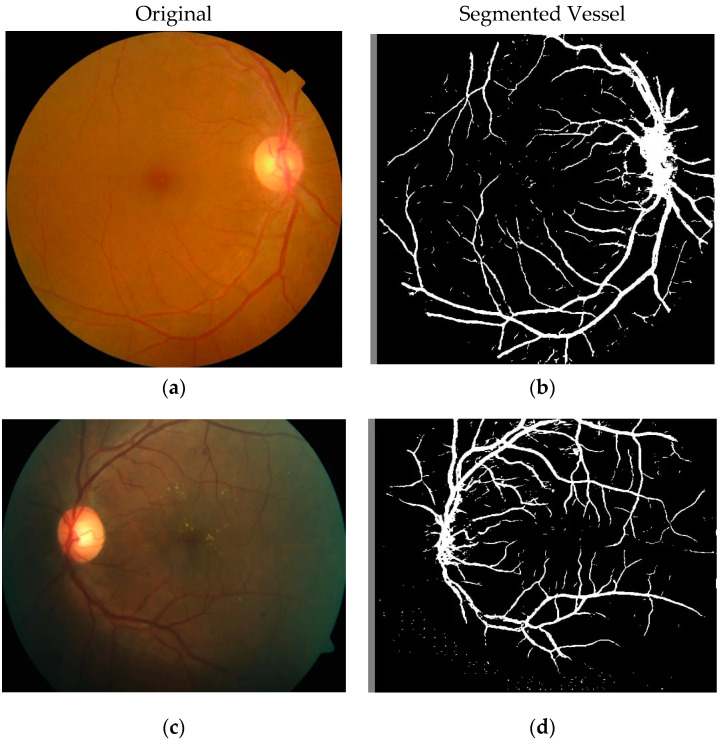
Image before segmentation and after segmentation stage. (**a**,**c**) Images before segmentation and (**b**,**d**) images after segmentation of their respective original images.

**Figure 2 bioengineering-10-00413-f002:**
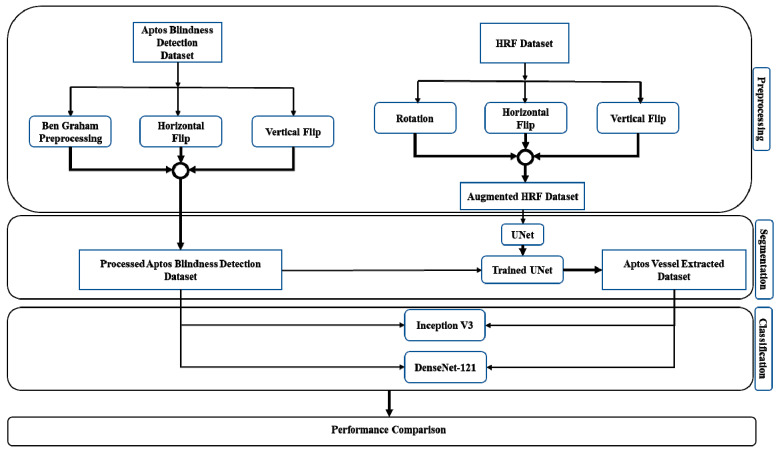
Block diagram of the proposed methodology.

**Figure 3 bioengineering-10-00413-f003:**
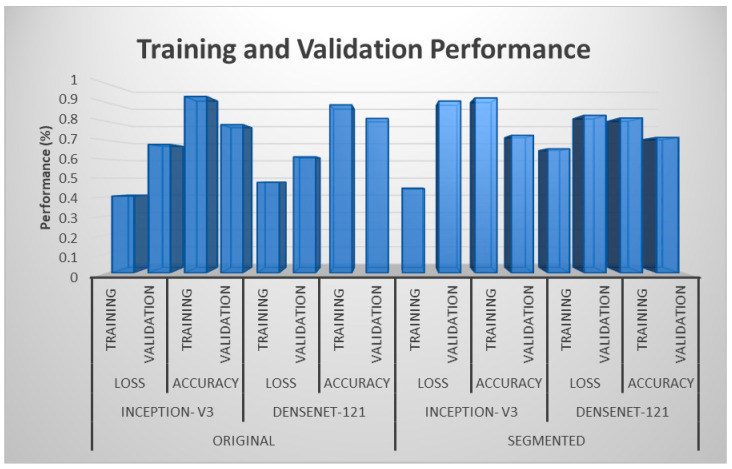
Training and validation performances for two different models.

**Figure 4 bioengineering-10-00413-f004:**
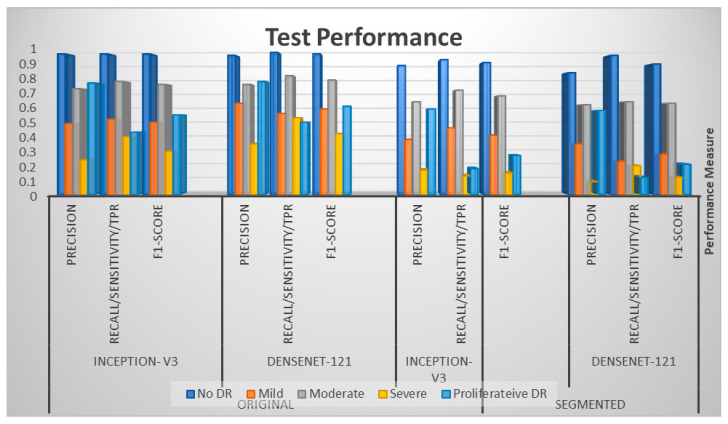
Test performances for Inception v3 and DenseNet-121.

**Figure 5 bioengineering-10-00413-f005:**
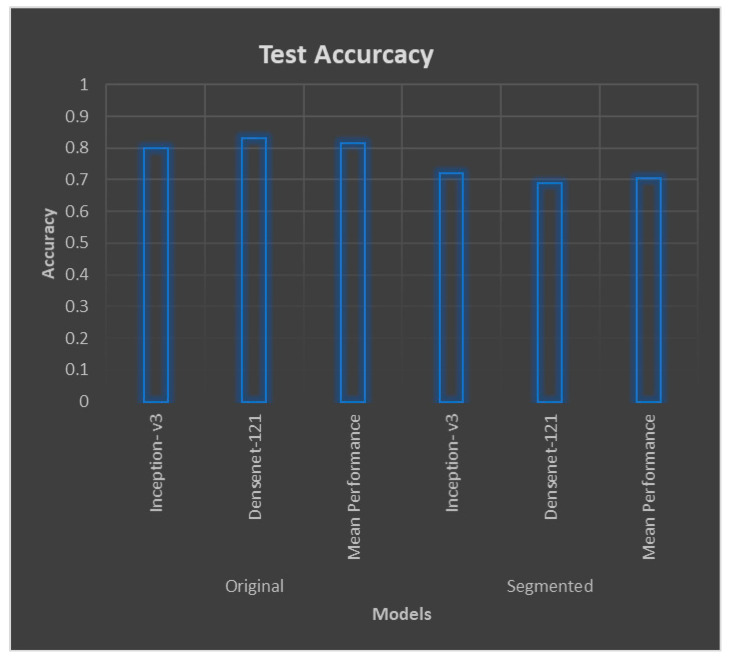
Mean test performances for original and segmented blood vessels.

**Figure 6 bioengineering-10-00413-f006:**
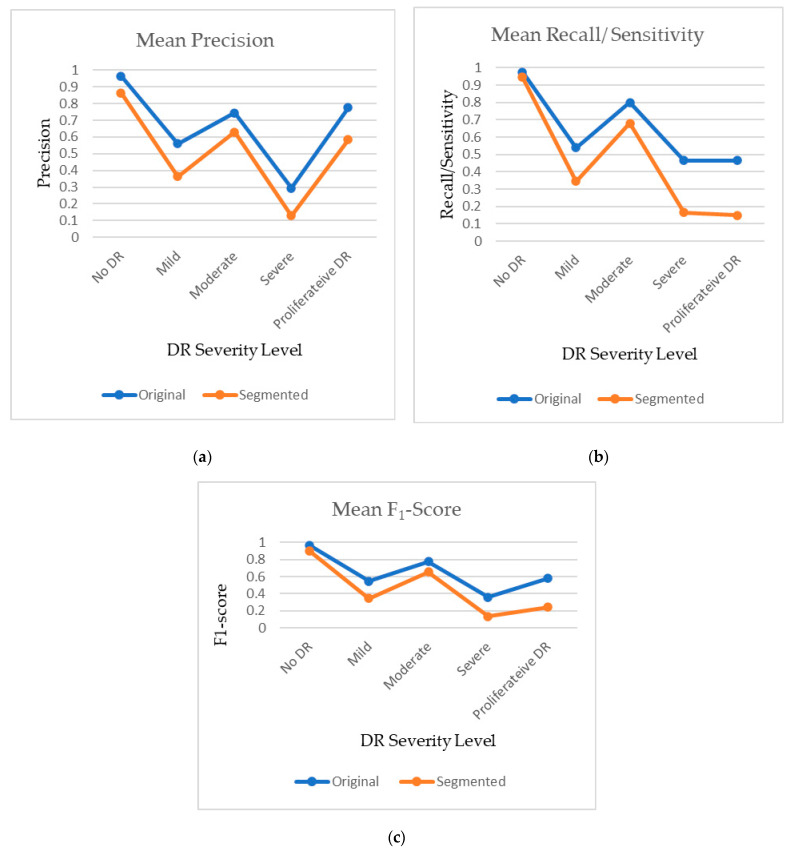
Plot of mean (**a**) precision, (**b**) recall, and (**c**) F_1_-Score for original and segmented datasets.

**Table 1 bioengineering-10-00413-t001:** Representation of traditional deep learning approaches for DR detection and classification over various datasets.

Ref.	DL Methods (Best Architectures)	Dataset	Performance Metrics			
			Accuracy	Sensitivity	Specificity	AUC
[15]	VGGNet	5-class (EyePACS)	95.68%	86.47%	97.43%	0.979
[16]	Custom CNN and Decision Tree	2-class (EyePACS) 2-class (Messidor2) 2-class (E-Ophtha)	--- --- ---	94% 90% 90%	98% 87% 94%	0.97 0.94 0.95
[8]	CNN (Inception v3)	Messidor-2 (1748) EyePACS-1 (9963)	---	96.1% 97.5%	93.9% 93.4%	---
[17]	CNN (ResNet50, Inception v3, InceptionResNet v2, Xception, and DenseNets)	Their own dataset (13,767)	96.5%	98.1%	98.9%	---
[18]	CNN (modified Alexnet)	Messidor (1190)	96.35%	92.35%	97.45%	---
[19]	CNN (VGGNet16, AlexNet, and custom CNN)	MESSIDOR (1200)	98.15%	98.94%	97.87%	
[20]	Fully CNN	STARE (20), HRF (45), DRIVE (40) and CHASE DB1 (28)	0.9628 0.9608 0.9634 0.9664	0.8090 0.7762 0.7941 0.7571	0.9770 0.9760 0.9870 0.9823	0.9801 0.9701 0.9787 0.9752
[21]	CNN (ResNet-101)	DRIVE (40)	0.951	0.793	0.974	0.9732
[22]	Custom CNN	5-class (IDRiD) 5-class (EyePACS)	91.3% 89.1%	--- ---	--- ---	--- ---
[23]	CNN (ResNet50)	Messidor (1200) IDRiD (516)	92.6% 65.1%	92% ---	--- ---	0.963 ---
[24]	CNN	HRF(45) and DRIVE(40)	93.94%			0.934
[25]	CNN (improved LeNet and U-net)	DIARETDB1 (89)		48.71%		0.4823
[26]	Ensemble learning	2-class (Private custom dataset)	88.21%	85.57%	90.85%	0.946
[27]	CNN	DRIVE(40) STARE(20) CHASE(28)	95.82% 96.72% 96.88%	79.96% 79.63% 80.03%	98.13% 98.63% 98.80%	98.30% 98.75% 98.94%

**Table 2 bioengineering-10-00413-t002:** Inception v3 and DenseNet-121 training parameter information.

Attribute	DenseNet-121	Inception v3
Optimizer	SGD	SGD
Base Learning Rate	1 × 10^−4^	1× 10^−4^
Momentum	0.9	0.9
Learning Decay Rate	1 × 10^−6^	1× 10^−6^
Train Batch Size	32	32
Trainable Parameters	7,217,541	22,294,181
Non-trainable Parameters	83,648	34,452
Total Parameters	7,301,189	22,328,613

**Table 3 bioengineering-10-00413-t003:** Test performance for original and segmented blood vessels.

	Original							Segmented								
	Inception v3			DenseNet-121				Inception v3				DenseNet-121				
STAGE	Precision	Recall/Sensitivity/TPR	F1-Score	Accuracy	Precision	Recall/Sensitivity/TPR	F1-Score	Accuracy	Precision	Recall/Sensitivity/TPR	F1-Score	Accuracy	Precision	Recall/Sensitivity/TPR	F1-Score	Accuracy
No DR	0.97	0.97	0.97	0.8	0.96	0.98	0.97	0.83	0.89	0.93	0.91	0.72	0.84	0.96	0.9	0.69
Mild	0.49	0.52	0.5		0.63	0.56	0.59		0.38	0.46	0.41		0.35	0.23	0.28	
Moderate	0.73	0.78	0.76		0.76	0.82	0.79		0.64	0.72	0.68		0.62	0.64	0.63	
Severe	0.24	0.4	0.3		0.35	0.53	0.42		0.17	0.13	0.15		0.09	0.2	0.12	
Proliferative DR	0.77	0.43	0.55		0.78	0.5	0.61		0.59	0.18	0.27		0.58	0.12	0.21	

**Table 4 bioengineering-10-00413-t004:** F_1_ Score, Precision, and Recall/Sensitivity/TPR for original and segmented blood vessels for DR severity levels.

Performance Metric	Type	No DR	Mild	Moderate	Severe	Proliferative DR
F_1_-Score	Original	0.97	0.545	0.775	0.36	0.58
	Segmented	0.905	0.345	0.655	0.135	0.24
Precision	Original	0.965	0.56	0.745	0.295	0.775
	Segmented	0.865	0.365	0.63	0.13	0.585
Recall/Sensitivity/TPR	Original	0.975	0.54	0.8	0.465	0.465
	Segmented	0.945	0.345	0.68	0.165	0.15

## Data Availability

The data used in this research are available at the following links: https://buffyhridoy.github.io/ (accessed on 19 June 2022). https://www.linkedin.com/in/mohammad-badhruddouza-khan-97b023195/ (accessed on 19 June 2022). https://www.researchgate.net/profile/Mohammad-Khan-353 (accessed on 19 June 2022).

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
