# Peer review of "Automated Diagnosis of Diabetic Retinopathy Using Deep Learning: On the Search of Segmented Retinal Blood Vessel Images for Better Performance"

_bioengineering, 2023, doi:10.3390/bioengineering10040413_

Round 1

Reviewer 1 Report

Thanks for submitting this work to MDPI Bioengineering. This is an interesting DL application forcusing on diabetic retinopathy diagnosis. There is no much study covering such interdisciplinary areas, so the apprearance of this work is timely. It aims to answer the question: which of the two DL models (InceptionV3 and Denstnet-121) performs better in this new scenario. 

The paper conducts comparative evaluation on two real data sets to evaluate the effectiveness of the two typical DL models and develops some insightful findings: the two DL models performs roughly equal towards the data sett; the segmented blood vessels are not considered as useful features to the DL application.

One minor thing I think the authors can add is the discussion on the choice of these models in practice. From my view, I feel DenseNet-121 should be preferred: regarding similar accuracy, DeseNet requires much less trainable parameters, which can save cost and enjoy higher training efficiency. The authors may conduct some netral discussion and draw some conclusions regarding the comparison of models.

Reviewer 2 Report

The paper provides a comparative study for detecting the various stages of diabetic retinopathy. The authors focused on two deep learning models, Inception V3 and DenseNet and used two different datasets to provide results.

Overall the paper has been well written and the conclusions build upon the results presented here. I have a couple of comments to make which I think will improve the paper further.

First, I think it is important in the introduction to highlight more the role of geometry and haemodynamics around diabetic retinopathy. This paper provides a good summary so it would be quite relevant:

Leontidis, G., Al-Diri, B. and Hunter, A., 2014. Diabetic retinopathy: current and future methods for early screening from a retinal hemodynamic and geometric approach. Expert Review of Ophthalmology9(5), pp.431-442.

Second, the rationale behind using Inception V3 and DenseNet is not clear, especially given that ResNet is the defacto CNN backbone for such models, as well as vision transformers

Third, figures 3, 4 and 5 might benefit from some changes as they look a bit outdated (up to you, I am not mandating it)

Fourth, a few typos, e.g. line 18 'Densetnet' should be 'Densenet', line 338 'Table.3' '.' is not needed, and a few more.

Finally, I find the outcome that segmentation does not offer a better outcome compared to coloured images. Perhaps draw some further conclusions as to why that might be the case, whether that is because we are losing the lesions, etc. maybe the paper I suggested above has some perspectives.

Author Response

Pease see the attached file.

Reviewer 3 Report

The manuscript entitled "Diagnosis of Diabetic Retinopathy using Deep Learning: Do Segmented Retinal Blood Vessels Produce Better Performance?" shows a study on the diagnosis of diabetic retinopathy using two methods of Deep Learning.

It provides information on the analysis of the original image and the segmented image, concluding that the segmented image does not provide greater improvements in diagnosis than what the original image provides. I would add that not only does it not provide improvements, but the results show that it is an approximation to the diagnosis with less precision than the original image and this should be made clear in the discussion and in the conclusion. Although this is good to know, it is not a relevant contribution to improving the diagnosis of diabetic retinopathy, so the impact of the results is relative.

In addition, the text has some inaccuracies and improvements that should be addressed:

-The title can do without the question as it confuses the reader. I suggest changing the title, focusing more clearly on the analysis of diagnostic methods. Something like this: Analysis of Deep Learning methods for Diagnosis of Diabetic Retinopathy

-Abstract: There is a redundant expression, please change "As a result" to another term so that it is not repeated in the summary so often.

-The introduction is adequate, but there is no clear limit that makes us understand that the materials and methods have begun since it is lost in the approach proposal. Please clarify when the materials and methods begin.

-In the same sense it is not clear when the results begin, but above all when they end. In section 5. Comparative analysis and discussion, I think it is necessary to separate the comparative analysis (which has a table and are results) from the discussion. They are mixed data that do not help the reader to understand the article properly. Please separate section 5 because it includes results in the discussion, and they are different sections.

-In the conclusion, please remove the first sentence (it is not a conclusion of this work) and focus on what has been obtained. Include here that the analysis of the segmented image has less resolution and precision than the analysis of the original image.

The manuscript need a major revision.

Reviewer 4 Report

1. There seems to be a lack of explanation regarding the datasets used in the study. It is unclear how the datasets were collected and what criteria were used for selection, which may affect the generalizability of the findings.

2. While the study provides useful insights into the efficacy of deep learning-based analysis for diabetic retinopathy diagnosis, it would be helpful to provide a discussion on the potential limitations and challenges of this approach in real-world clinical settings.

3. The authors need to consider comparing their proposed method and similar approaches in the introduction. Specifically, it would be helpful to discuss and compare their approach with relevant papers in the field, such as "Artificial Intelligence and COVID-19: Deep Learning Approaches for Diagnosis and Treatment", "A conceptual deep learning framework for COVID-19 drug discovery" and "Cancer Digital Twins in Metaverse". 

4. The statement that segmented vessels do not add much utility to deep learning-based analysis could benefit from further clarification. It is unclear if this conclusion applies to all cases of diabetic retinopathy or if it is limited to the specific datasets used in the study.

5. The content would benefit from more detailed information on the methodology used in the study, including the parameters used in the deep learning algorithms and the criteria for evaluation.

6. The implications of the study's findings for clinical practice could be elaborated further. How can this information be used to improve screening and diagnosis of diabetic retinopathy in real-world settings? What are the potential benefits and challenges of implementing deep learning-based approaches in clinical practice?

Round 2

Reviewer 3 Report

The authors have followed the suggestions and now the manuscript is ready to be published.

Congratulations!!1

Author Response

The authors are very glad to know that the reviewer is satisfied with the authors response to the reviewer's comments. We are also happy to know that the reviewer has given his/her consent to the Editor to publish our manuscript in this journal.

Finally, we are thankful to the reviewer for congratulating us !

M. A. Rashid

!!1

Reviewer 4 Report

Except for comments 4 and 5, none of the comments have been addressed. 
